# Is Endothelial Activation a Critical Event in Thrombotic Thrombocytopenic Purpura?

**DOI:** 10.3390/jcm12030758

**Published:** 2023-01-18

**Authors:** Raphael Cauchois, Romain Muller, Marie Lagarde, Françoise Dignat-George, Edwige Tellier, Gilles Kaplanski

**Affiliations:** 1Aix Marseille University, Assistance Publique Hôpitaux de Marseille, INSERM, INRAE, C2VN, CHU Conception, Internal Medicine and Clinical Immunology, 13005 Marseille, France; 2French Reference Center for Thrombotic Microangiopathies, 75571 Paris, France; 3Aix Marseille University, INSERM, INRAE, C2VN, 13005 Marseille, France; 4Aix Marseille University, Assistance Publique Hôpitaux de Marseille, INSERM, INRAE, C2VN, CHU Conception, Hematology Laboratory, 13005 Marseille, France

**Keywords:** thrombotic thrombocytopenic purpura, endothelial cells, Weibel–Palade bodies

## Abstract

Thrombotic thrombocytopenic purpura (TTP) is a severe thrombotic microangiopathy. The current pathophysiologic paradigm suggests that the ADAMTS13 deficiency leads to Ultra Large-Von Willebrand Factor multimers accumulation with generation of disseminated microthrombi. Nevertheless, the role of endothelial cells in this pathology remains an issue. In this review, we discuss the various clinical, in vitro and in vivo experimental data that support the important role of the endothelium in this pathology, suggesting that ADAMTS13 deficiency may be a necessary but not sufficient condition to induce TTP. The “second hit” model suggests that in TTP, in addition to ADAMTS13 deficiency, endogenous or exogenous factors induce endothelial activation affecting mainly microvascular cells. This leads to Weibel–Palade bodies degranulation, resulting in UL-VWF accumulation in microcirculation. This endothelial activation seems to be worsened by various amplification loops, such as the complement system, nucleosomes and free heme.

## 1. Introduction

Thrombotic thrombocytopenic purpura (TTP) is a rare and severe disease belonging to the thrombotic microangiopathies disorders and described for the first time in 1924 by Eli Moschcowitz. It is characterized by uncontrolled platelet aggregation and adhesion which will form microthrombi resulting in the clinical syndrome of TTP. Clinical presentation includes thrombocytopenia, mechanical hemolysis and organ damage. It can be fatal without prompt diagnosis and appropriate treatment [1]. Knowledge about TTP pathophysiology has considerably improved in recent decades: in 1982 Joel L Moake identified uncleaved von Willebrand Factor (VWF) multimers in patients with chronic relapsing TTP [2]. Nineteen years later, ADAMTS13 (A Disintegrin and Metalloproteinase with Thrombospondin-1 motifs, 13th member of the family) was related to the loss of function of the VWF multimers-cleaving metalloproteinase [3,4]. The severe ADAMTS-13 deficiency can be inherited as in congenital TTP (Upshaw–Schulmann syndrome [5]) or more commonly acquired due to inhibitory auto-antibodies against ADAMTS13 (i-TTP, for immune-mediated TTP) [6,7].

The current pathophysiological paradigm suggests that the ADAMTS13 deficiency leads to Ultra Large-VWF multimers (UL-VWF) accumulation on endothelial cells [8]. Flowing blood applies a tensile force that “unfolds” UL-VWF, which, therefore, form pro-thrombotic strings into micro vessels. They induce a massive platelet adhesion and aggregation with rapid generation of disseminated microthrombi, leading to the thrombotic microangiopathies characteristics triad: (i) organ ischemia, (ii) profound thrombocytopenia and (iii) hemolytic anemia. Organ ischemia affect mainly central nervous system but also heart, digestive tract and occasionally kidney. Thrombocytopenia (often < 30 G/L) is often associated with hemorrhagic risk because of platelet’s consumption in thrombi. Furthermore, recent research has shown that thrombocytopenia is aggravated by insufficient bone marrow production of young platelets [9,10]. Anemia is accompanied with schistocytes on blood smear due to mechanical erythrocyte fragmentation on thrombi.

Due to its clinical severity, patients with i-TTP require urgent treatment in intensive care unit. Daily therapeutic plasma exchanges (TPE), commonly used since the 80’s, represent the cornerstone of acute phase management and have significantly improved outcomes by drastically reducing mortality from 90% to 20% two weeks after diagnosis [11]. TPE are very effective by providing exogenous ADAMTS13 contained in donor’s plasma and by removing anti-ADAMTS13 antibodies. Due to the autoimmune pathophysiology, corticosteroids are systematically associated with TPE during the acute phase. Rituximab, an anti-CD20 monoclonal antibody is another frontline therapy [12]. More recently, Caplacizumab, a nanobody inhibiting VWF-platelet interaction, provides an interesting protective effect during the acute phase [13,14,15]. Despite those recent therapeutic improvements, TTP remains a life-threatening disease with high morbidity [16] and relapse risk for patients who survive [17].

In addition to the central role of ADAMTS13 deficiency in TTP pathogenesis, the endothelial cells, which represent the main UL-VWF storage, have been long recognized as major actors in TMA and in TTP particularly [18,19]. Indeed, many TTP related studies have described endothelial damages, especially histopathological such as swelling/necrosis and subendothelial hyaline deposits, biomarkers association with endothelial activation and frequent environmental triggers. The existence of an endothelial activation during TTP acute phase is accepted by the scientific community. However, one question remains: is the endothelial activation a consequence of the micro-occlusive disease [20], i.e., an epiphenomenon, or is it a key initiating event that precipitates an individual susceptibility into an acute episode? Many clinical and experimental data support the hypothesis that endothelial activation is essential to the pathogenesis. This activation represents a “second hit” that induces a massive UL-VWF release into the microcirculation. 

After a succinct presentation of the endothelium and the Weibel–Palade bodies, we will present elements that support or not this hypothesis. Then, we will discuss the nature of the suspected triggers and the existence of amplification loops increasing the severity degree of TTP.

## 2. The Endothelium

### 2.1. Endothelial Cells

The endothelium is a cellular monolayer that covers the luminal surface of the vascular tree [21]. Endothelial cells (ECs) are polarized through their actin cytoskeleton and constitute a dynamic interface between flowing blood (apical pole) and sub-endothelial tissues (basal pole). They play a central role in the regulation of major physiologic functions, such as hemostasis, vascular permeability, cellular and nutrient trafficking, inflammation, innate and adaptive immunity, vascular tone and angiogenesis.

Structure and function heterogeneity constitute key features of the endothelium. Depending on the vascular territory and vascular bed [22,23], endothelium presents a spatial phenotypic diversity. The morphology, secretory repertory and behavior of ECs differ between arteries and veins, and between microvascular (vessel size < 300 µm) and macrovascular (>300 µm) territories. Endothelial heterogeneity is also temporal: EC phenotype shows a high plasticity degree and phenotypical changes can occur in response to many physiological or pathological conditions, such as biomechanical signals, pH variations, hypoxemia, soluble mediators or cell–cell interactions.

### 2.2. Endothelial Activation

Quiescent ECs display a thromboresistant (anticoagulant and antiaggregant) and anti-inflammatory phenotype. They also protect flowing blood from the highly prothrombotic subendothelial matrix [24]. These features prevent spontaneous thrombi formation in blood vessels and result from antithrombotic molecules expression: TFPI (Tissue Factor Pathway Inhibitor), existing in transmembrane and soluble isoform, TM (Thrombomodulin), EPCR (Endothelial Protein C Receptor), t-PA (tissue-type Plasminogen Activator), NO (Nitric Oxide) and PGI2 (Prostacyclin). When exposed to various stress signals, such as cytokines, toxins, disturbed shear stress, etc., ECs acquire an “activated phenotype” which is prothrombotic (Figure 1). Consequences of this activation are multiple: VWF-multimers released from the Weibel–Palade bodies, reduced expression of the above-mentioned thromboresistant molecule and increased expression of Tissue Factor (TF), the main activator of the extrinsic coagulation pathway and Plasminogen-Activator Inhibitor-1 (PAI-1). Furthermore, activated ECs also acquire a pro-inflammatory phenotype characterized by leukocyte receptor expression, such as P-selectin, and an increase in intercellular permeability facilitating leukocyte trafficking.

Moreover, endothelial activation triggers cell vesiculation [25], leading to endothelial microvesicles (EMVs) generation and release in blood flow. These extracellular vesicles have a size ranging from 0.1 to 1 µm. EMV possess an essential role in intercellular communication. Their biogenesis occurs via the plasma membrane blebbing of ECs with whom they share common characteristic phenotypic features. EMVs have procoagulant properties because of the phosphatidylserine and TF, but they also seem to be involved in fibrinolysis with uPA or tPA [26]. They are generally pro-inflammatory and partly responsible for the dysregulation of vascular tone in endothelial dysfunction. Thus, EMVs, considered only as biomarkers for a long time, are likely to be also biologic effectors that actively participate to the endothelial activation consequences. 

As pointed out by Roumenina et al. [27], ECs constitute all together a source, a barrier in well-regulated conditions and a target in pathologic conditions, for hemostasis and complement system actors. They form a “quiescent endothelium” different of the “activated endothelium” under regulated activation or in response to various stimuli. This activated endothelium can also lead to the “saturated endothelium” when ECs regulation capacity are exceeded. Then, two situations can occur: (1) the endothelial activation is too intense and it exceeds cells physiologic regulation potential (e.g., during septic shock) or (2) an acquired or innate susceptibility (e.g., ADAMTS13 deficiency) leads to disastrous repercussions: ECs are then the main target and actor of amplification loops.

A key point is that endothelial activation is a dynamic continuum between a quiescent and an activated phenotype. There is also a spatial heterogeneity of phenotypic features that would help us to distinguish between these two phenotypes [28] and a modulation of cytokines effects on ECs depending on the vascular site.

Finally, two types of endothelial activation have been usually described based on inflammatory context [29]. ECs type 1 activation typically occurs within few minutes. It does not involve gene expression modulation and is mediated by G-protein-coupled-receptors (GPCRs) leading to an intracellular calcium (Ca^2+^) influx. ECs type 2 activation is mediated by a more sustained signal. It involves gene expression regulation. Tumor-Necrosis Factor (TNFα) and Interleukin-1 are the main mediators of this response. These signaling leads to the nuclear translocation of the transcription factor NFκB and AP-1, resulting in cytokine and chemokine production. Nevertheless, those two types of activation are mutually dependent and many crosstalk exist.

### 2.3. Weibel–Palade Bodies

Weibel–Palade Bodies (WPBs) are endothelial-specific storage organelles and their composition is closely linked to the physiological regulatory functions of the endothelium [30]. These cigar-shaped granules come from the Golgi apparatus. Their biogenesis is complex and is not fully understood but seems to depend on VWF multimerization process [31]. WPBs contain UL-VWF, which has a major role in hemostatic function. They also contain P-selectin, an important leukocyte adhesion molecule, Interleukin-8 (cytokine), eotaxtin-3 (chemokine), Endothelin-1 (vasoconstrictor) and Angiopoietin-2 involved in angiogenesis. Other molecules have been identified in CWP, such as the Complement Factor H, Osteoprotegerin, t-PA, etc. However, this list is not exhaustive because of the spatial and temporal WPB heterogeneity similar to the ECs heterogeneity previously described [30,31,32].

The endothelial activation leads to WPBs fusion with the apical plasma membrane and the subsequent release of their content in the blood circulation. Two different pathways of WPB agonists-induced degranulation are described [32]:–The Ca^2+^-mediated pathway: agonists, such as thrombin and histamine activate G_q_PCRs and G_i_PCRs, leading to the activation of phospholipase Cβ and the formation of inositol triphosphate (IP3). The fixation of IP3 on the endoplasmic reticulum membrane IP-3 receptor generates a Ca^2+^ intracellular influx which triggers the degranulation of the cortical pool of WPBs. It also increases vascular permeability involving VE-Cadherin phosphorylation and myosin light-chain phosphorylation and participates in the vesiculation process [33].–The cAMP-mediated pathway (cyclic adenosine monophosphate): other agonists, such as epinephrine and serotonin activate G_s_PCRs, inducing an increase in cAMP intracellular level. This results in the activation of the PKA (protein kinase A) that triggers a lower degranulation of WPB microtubular pool. In contrast to the Ca^2+^-mediated pathway, cAMP-mediated pathway is associated with a decrease in vascular permeability.

### 2.4. From Endothelial Activation to Endothelial Dysfunction

The endothelial activation is a well-regulated physiological stress-response. Nevertheless, it may have adverse impacts when it becomes chronic or uncontrolled. Indeed, the chronic EC activation leads to endothelial dysfunction affecting vascular tone, hemostasis, and inflammation. The term of endothelial dysfunction was initially used in the context of atherosclerosis [34,35], to describe the impaired NO-dependent vasodilatation to various biomechanical or chemical stimuli, such as an increase in blood flow or acetylcholine. However, a chronic EC activation also leads to an impaired regulation of other functions, resulting in a prothrombotic and pro-inflammatory phenotype. Chronic endothelial dysfunction involves a reduction in NO bioavailability and an oxidative stress which takes part in eNOS (endothelial NO-synthase) uncoupling. Disturbed flow can actively participate in endothelial dysfunction, inducing epigenetic modification in ECs [36,37]. Endothelial dysfunction is also associated with a greater endothelial senescence and apoptosis and it is a marker, and, most probably, also an actor of the global cardiovascular risk.

## 3. Evidence for Endothelial Activation in TTP

One of the key features of TTP pathogenesis is UL-VWF accumulation. The endothelial activation leads to UL-VWF massive release through CWPs degranulation. This phenomenon is supposed to be the trigger for TTP crisis when occurring in an ADAMTS13 deficiency context. In this part, we discuss clinical and experimental data that support or contradict this hypothesis.

### 3.1. Clinical Points

Patients with not treated hereditary TTP have a null or very low ADAMTS13 activity from birth. However, the first TTP crisis can occur at adulthood, especially during pregnancy [8,38,39]. Similarly, patients with i-TTP may have a very low ADAMTS13 activity (<5%) without displaying biological or clinical features of TMA [40,41]. These data suggest that ADAMTS13 deficiency is a necessary but not sufficient condition to induce TTP. Furthermore, an infectious episode is frequently reported few weeks before the crisis supporting the hypothesis of an environmental triggering factor. 

### 3.2. Endothelial Exploration in Humans in TTP

Measurements of ECs activation degrees are possible by various in vitro studies, such as morphologic analysis, phenotypic markers analysis or functional tests. However, in clinical practice, evaluation of ECs activation degree is not easy. Nevertheless, some investigative methods exist soluble and cellular biomarkers analysis and histological studies.

Biomarkers analysis is a systemic approach of endothelial activation in clinical practice [42,43,44]. Classically plasmatic biomarkers analyzed are hemostatic biomarkers (soluble TM, soluble TF, VWF, t-PA and PAI-1), cytokines and soluble forms of adhesion proteins, such as VCAM, ICAM, E-selectin and P-selectin. High levels of soluble biomarkers of endothelial injury during the acute phase of TTP has been highlighted for a long time. Takahashi et al. reported in 1991 that plasmatic TM concentration was higher in TTP patients compared to healthy subjects, and decreased during remission [45]. Moreover, Mori et al. reported that plasmatic TM was associated with mortality in TTP patients, suggesting that TM plasmatic level could constitute a prognosis factor [46]. Soluble P-selectin [47,48], soluble t-PA [49], soluble PAI-1 and VWF:Ag (VWF antigen) have been proposed as EC activation markers in TTP patients plasma. Soluble P-selectin is also correlated to usual markers of TTP activity (platelet and LDH levels) [50]. Some authors support that endothelial activation leads to a procoagulant state [51] without high fibrinolysis level [52]. This idea is based on hemostatic parameters, such as sTFPI/sTF ratio, t-PA and PAI-1 analysis, although t-PA data differ between studies [49,52]. These parameters seem today obsolete in TTP context and their relevance questionable. Indeed, platelet-rich and fibrin-poor thrombi are the hallmark of the disease [53], suggesting a modest involvement of the coagulation system in TTP. Some data suggest that fibrinolytic system induced by ECs hypoxia may be a bypass pathway for VWF multimers cleavage [54]. Then, an alteration of ECs fibrinolytic potential has been proposed as a critical event for TTP crisis [55]. Van Mourik et al. proposed to analyze the VWF:Ag/VWF-propeptide ratio to distinguish between an acute or a chronic endothelial activation [56]. Indeed, the secretions of VWF:Ag and its propeptide are equimolar but their half-life are different: the propeptide is cleared from the circulation five times faster than the VWF:Ag. However, this ratio may be difficult to interpret in the TTP context: the ADAMTS13 deficiency modifies the half-life of VWF and the TPE provides de novo ADAMTS13. Furthermore, the dynamics for reconstitution of WPBs is not fully understood and a high level of circulating UL-VWF is not associated with TTP crisis [57]. The clinical relevance of these parameters is nowadays questionable [43].

Cellular biomarkers have been developed more recently. As mentioned, ECs activation and dysfunction are associated with EMVs production, but also with the detachment of ECs from the basal membrane. This results in generation of circulating endothelial cells (CEC). EMVs and CECs levels, which reflect endothelial injury, have been associated with an increase in global cardiovascular risk and may be poor prognosis factors for many diseases [25,44]. In addition to being biomarkers, they probably actively participate in vascular damage through vectorization of deleterious signals. Jimenez et al. were the first to document EMVs in TTP. They demonstrated that TTP plasmas induced a procoagulant EMVs generation from cultured brain and renal microvascular ECs (MVECs) [58]. They also shown elevated concentrations of EMVs in TTP patients during acute phase normalized during remission [59], shortly before LDH and platelet normalization. On the other hand, CEC were identified in 1993 in a TTP patient [60] and this was confirmed by Widemann et al. in a prospective multicentric study on 22 patients: CECs were elevated during acute phase of the TTP and normalized during remission and high CEC level at diagnosis was associated with clinical severity [50].

Biopsies with morphologic analysis and immuno-histological investigations of endothelium are rarely practiced. The procedure is invasive, and the subsequent analysis remains focal. However, histological features of endothelial lesions are frequently observed in other thrombotic microangiopathies, especially on renal biopsy. Above-mentioned data support that endothelial activation occurs in TTP and may be a prognosis factor. However, it remains to determine if ECs activation is a cause or a consequence of thrombi formation. Dang et al. provide responses elements [61]: they analyzed 8 spleens from TTP patients and highlighted thrombi and EC apoptosis overexpressed Fas as compared to control spleens. Interestingly, many apoptotic cells did not colocalize with thrombi, suggesting that ECs apoptosis precedes thrombi formation. However, as noticed by Jimenez [62], a percentage of ECs apoptosis should be interpreted with analysis of EC activation degree as Fas is also an ECs activation marker through NFκB pathway.

### 3.3. In Vitro Experimental Data

In 1982, Burns et al. reported that serum from TTP patients induced endothelial lesions in vitro as compared to serum from healthy volunteers [63]. They highlighted a pathogenic effect of the IgG fraction that fixed HUVECs (Human Umbilical Vein ECs) and induced ultrastructural lesions. Many other studies subsequently demonstrated a pathogenic effect of TTP plasmas on ECs. Laurence et al. reported the pro-apoptotic effect of 4 TTP plasmas on microvascular ECs, but not on macrovascular ECs. One of which was associated with HIV infection. This was not observed with plasmas from patients with disseminated intravascular coagulation or from asymptomatic HIV patients [64]. This pro-apoptotic effect seemed to be Fas-mediated and independent from TNFα or CD36 pathway. Mitra et al. then showed that this pro-apoptotic effect was induced with plasma from acute TTP patients, but not with plasma from patients in remission and only in restricted lineage of microvascular ECs [65]. Thus, Human Microvascular Endothelial Cells (HMVECs) from hepatic and pulmonary lineage were not affected, liver and lung being interestingly generally spared during TTP crisis. Jimenez et al. highlighted a mainly activator effect of TTP plasma on HMVECs, without a clear pro-apoptotic effect, based on phenotypic features of EMVs generated [59]. They also showed that EMVs generated in vitro carried VWF and induced platelet aggregation [66]. Discordances between these studies may be at least partly explained by the use of different experimental methods [62,67]. Finally, our team recently showed that plasmas from acute TTP patients induced a calcium and IgG WPB degranulation in vitro with a strong correlation with the pathology severity [68].

### 3.4. In Vivo Experimental Data

Motto et al. were the first to generate a TTP mice model, knocking-out ADAMTS13 gene on a mixed strain C57BL6/J–128X1/SvJ background [69]. Despite the loose of ADAMTS13 activity, mice did not present any clinical or biological TMA feature. These results were confirmed by Banno et al. [70]. Then, the mice were backcrossed on the CASA/Rk background which confers higher VWF levels. Consequently, a prothrombotic state was observed, with mild thrombocytopenia and a survey slightly decreased. TTP was finally induced through intravenous Shigatoxin administration. Shigatoxin is a bacterial toxin mainly produced by certain strains of Escherichia Coli and responsible for typical Hemolytic and Uremic Syndrome (HUS), another TMA that principally affects the kidney. Shigatoxin exists on two isoforms and is composed of two subunits: the A-subunit induces a ribotoxic stress leading to EC death and the B-subunit induces the WPBs degranulation. This B-subunit is sufficient alone to induce TTP in this model [71]. Interestingly, double knock-out mice for ADAMTS13 and for VWF were protected from shigatoxin deleterious action [72], confirming that UL-VWF release is a key point of pathophysiology of TTP.

Feys et al. developed a TTP baboon model [73]. Authors generated murine blocking monoclonal antibodies (mAb) against human recombinant ADAMTS13. They injected this mAb to 6 healthy baboons. ADAMTS13 activity was lost and biological and histological features of TTP present, without clinical repercussion. Authors concluded that, contrary to observations in the mice model, an additional trigger was not necessary to induce TTP in baboons. The differences observed between the mice and the baboon models could be related to supplying protease in mice, such as plasmin, enough in case of isolated ADAMTS13 deficiency, but exceeded in case of massive endothelial activation. Nevertheless, ECs activation was not assessed in the baboon model and a direct effect of anti-ADAMTS13 antibodies on endothelial cells could not be excluded [74].

Le Besnerais et al. examined endothelial injury in another model of TTP, in which PTT was induced with administration of recombinant VWF, including UL-VWF, in ADAMTS13-deficient mice [74]. A systolic dysfunction was highlighted with a decrease in myocardial perfusion on magnetic resonance imaging, associated with an alteration of NO-mediated relaxation response in coronary and mesenteric arteries. These results reflected early endothelial dysfunction. Cardiac ECs presented a globally proadhesive state with overexpression of VCAM and E-Selectin and a pro-oxidative state.

Recently, Zheng et al. generated ADAMTS13 −/− zebrafish which exhibited spontaneously a mild thrombocytopenia with increased fragmentation of red blood cells and presented a prothrombotic state [75]. TTP was induced through a Lysine-rich histones injection, known to be WPBs calcium-dependent degranulation inducers [76]. Thrombocytopenia was more severe and microvascular VWF-rich microthrombi in the ADAMTS13 −/− zebrafish group more present compared to the wild type group. However, the Kaplan-Meyer survival analysis indicated a 2 week mortality rate after the histone challenge around 25% in the ADAMTS13 −/− group, which corresponds to a TTP model much less severe than in humans. The prothrombotic features (spontaneous or histones induced) of ADAMTS13 −/− zebrafish were—as expected—completely rescued in case of double knock out ADAMTS13 −/− and VWF −/−.

Other TTP rodent models have been developed, especially some auto-immune models [77,78,79]. In all of those, TTP induction needed recombinant VWF or Shigatoxin administration.

## 4. What Are the Suspected Triggers for the Second Hit Hypothesis?

Pregnancy is a common trigger of TTP crisis [80,81]. As mentioned above, patients with congenital ADAMTS13 deficiency may be asymptomatic and, thus, undiagnosed for decades until a first TTP crisis. Pregnancy is associated with an ad hoc prothrombotic state, linked to an increase in coagulation factors, circulating VWF levels and a concomitant low decrease in ADAMTS13 activity [82,83,84]. These additional variations may precipitate the TTP crisis.

Infections have often been suspected to be EC activation triggers in TMA disorders. Indeed, a flu-like episode is frequently reported during prodromes. In the literature, many cases reported TTP crisis occurring subsequently to infections, especially viral [85,86]. A prospective study conducted by the French Reference Center for Thrombotic Microangiopathies highlighted an infectious event within 2 weeks before the diagnosis in 41% of the 280 patients [87]. Different mechanisms may establish the link between the infectious process and TTP crisis:–Some viruses, such as the Cytomegalovirus (CMV), have a tropism for ECs and can directly activate them [88].–The cytokines secreted in response to the infectious process, especially γ-interferon, TNFα and Interleukin-8, may act as CWP degranulation inductors [89]. Type 1 interferons (α and β) may induce TMA, as shown in a murine model [90]. Furthermore, γ-interferon and TNFα are cytokines that downregulate *Adamts13* gene expression in the liver [91].–TLR-9 (Toll-like receptor-9) is a molecule belonging to innate immunity expressed on neutrophils and ECs which recognizes DNA present in bacteria and viruses. TLR-9 polymorphisms have been suspected to be a genetic risk factor of TTP crisis [87].

In addition, HIV can lead to two distinct TMA [92]. HIV infection at the AIDS stage may induce a non-specific TMA with ADAMTS13 activity > 5%. This TMA has a very poor prognosis and may be mediated by angio-invasive infections in the context of major immunosuppression. However, HIV can also lead to i-TTP, with a better prognosis. Presence of p24 antigen in ECs from bone marrow has been reported in one patient [93], but this case seems isolated and pathogenesis of HIV-induced i-TTP is globally unknown.

Finally, many cases of TTP following COVID-19 have been reported [94]. SARS-CoV-2 is now well known as an endothelial activator [95] and may therefore precipitate TTP in susceptible individuals. ACE2, the main receptor of SARS-CoV-2, has been identified on endothelial cells, but other receptors also seem involved in viral entry (neuropilin 1, CD147) [96]. Viral entry in endothelial cells would even not be necessary, indeed Lei et al. described that proteins S and N would be sufficient to cause endothelial dysfunction [97]. Furthermore, endothelial damages in COVID-19 are also secondary to infection of neighboring cells and of hyper-inflammatory syndrome [98].

Many drugs have been incriminated for the development of TMA, such as calcineurin inhibitors, mitomycin, gemcitabine, anti-VEGF agents and quinine [8]. Generally, these drug-induced TMA are not associated with a profound ADAMTS13 activity deficiency. Many mechanisms are suspected and they overall involve ECs aggression, inducing an activated phenotype of ECs. For example, calcineurin inhibitors induce a prothrombotic and proinflammatory state with a decrease in NO and PGI2 production and an increase in endothelin-1 and thromboxane A2 synthesis [19]. Ticlopidine (a thienopyridine platelet-antagonist) seems to be an exception, as it induces TTP within 2 to 12 weeks after introduction of the drug [99]. Today, pathogenesis of this induced auto-immunity is very uncertain. Interestingly, Mauro et al. showed that pharmacological doses of ticlopidine induced in vitro EC apoptosis with the same restricted lineage affected as discussed above: macrovascular ECs and hepatic and pulmonary microvascular ECs were spared from this pro-apoptotic response. This study relied ticlopidine induced in vitro EC apoptosis to an alteration of ECs interaction with subendothelial matrix [100].

As previously mentioned, IgG fraction has been incriminated for long to explain endothelial aggression occurring in TTP. The mechanisms involved in auto-immunity in TTP remains unknown. The loss of tolerance to ADAMTS13 may be linked to a genetic predisposition involving the major histocompatibility complex class II (HLA DRB1*04 as protective and HLA DRB1*11 and DQB1*03 as predisposing) [101,102]. Interestingly, peptides from CUB2 domain of ADAMTS13 were presented on HLA DRB1*11 [103] and there is some evidence of involvement of CUB2 domain-reactive CD4+ T Cells in i-TTP [104]. Many studies highlighted the presence of antibodies against ECs (AECA) in plasmas of TTP patients. These antibodies have the property to induce CWP degranulation in vitro [105]. Even if the targeted epitope is not always known, two studies reported a significant proportion of TTP patients with anti-CD36 antibodies [106,107]. CD36 is an antigen present on microvascular ECs but not on macrovascular ECs. In 2000, Praprotnik et al. proposed a pathophysiology including two auto-immune hits: AECA would activate microvascular ECs, inducing a massive UL-VWF release, that, therefore, accumulate into microcirculation because of the immune-mediated ADAMTS13 deficiency [105]. Nevertheless, the proportion of patients with AECA is a controversial issue and a significant part of these antibodies are thought to target the major histocompatibility complex (MHC) [108]. Indeed, in allogeneic kidney transplantation, donor specific antibodies may induce ECs aggression, leading to acute or chronic humoral graft rejection with TMA histologic features [109]. Moreover, Ren et al. showed that administration of xenogeneic AECA to rats led to dose-dependent and complement-mediated TMA features [110]. We also demonstrated a potential role of ADAMTS13 antibodies: anti-ADAMTS13 monoclonal antibodies purified from iTTP patients B cells induced an ECs activation calcium dependent leading to a degranulation of WPBs [68]. 

Surgery or endovascular procedures have been described as trigger of TTP [111]. Endothelial activators (alpha-thrombin, nucleosomes, reactive oxygen species, etc.) generation during ischemia-reperfusion could lead to endothelial cell activation with vWF release [112].

Recently, TMA were observed after a snake bite in Sri Lanka, especially by *Hypnale hypnale*, as known as the hump-nosed viper [113]. Venom analysis highlights the presence of phospholipase A2, serine-proteases, metalloproteinases and of thrombin-like enzyme [114,115]. Effects of these venoms on ECs may be an interesting track to explore.

## 5. Amplification Loops of Endothelial Aggression

TTP is characterized by ADAMTS13 deficiency and the accumulation of UL-VWF following endothelial activation. This leads to the involvement of other actors that maintain and aggravate endothelial damage leading to a “saturated endothelium” stage. These amplification loops involve the alternative complement pathway, hemolysis and nucleosomes and are shown in Figure 2.

### 5.1. The Complement System

The complement system is an archaic defense system belonging to innate immunity. It is composed of a large number of proteins acting through a highly regulated cascade activation [116]. Complement activation leads to three main consequences: (i) pathogen agents’ lysis through the action of C5b9, the membrane attack complex; (ii) apoptotic cells clearance through opsonization by C3b; and (iii) amplification of inflammatory response, through the action of C3a and C5a anaphylatoxins. The activation cascade can be initiated through a classical, lectin or alternative pathway. These pathways converge in a common terminal effector, the C5b9. The alternative pathway of the complement (APC) is characterized by a permanent low activation degree through spontaneous C3 hydrolysis. It needs to be regulated by soluble inhibitors, such as complement factor H and factor I, respectively, CFH and CFI and membranous inhibitors, such as Membranous Cofactor. 

The complement system is involved in many thrombotic microangiopathies. Atypical HUS is a TMA affecting mainly kidneys and is a well-known “complementopathy” model [117]. Up to 60% of patients have an identified anomaly affecting APC: mutations leading to a loss of function affecting regulators (CFH, CFI, MCP and others) or an inhibitory antibody against CFH; mutations leading to a gain of function affecting activators C3 and Complement Factor B (CFB). This dysregulated activation of APC leads to ECs aggression, especially in the presence of an additional trigger, such as glycocalyx alteration or free heme release. This induces a strongly pro-thrombotic and pro-inflammatory endothelial phenotype with microthrombi formation, especially in glomerular endothelium. Eculizumab is a monoclonal antibody against the C5 fraction of complement targeting the common terminal pathway. It has dramatically improved outcomes of complement-mediated HUS. Other data support the complement involvement in pathogenesis of typical HUS [118,119], preeclampsia [120] and allogeneic hematopoietic stem-cell transplantation (HSCT) [121,122,123] related TMA. Thus, complement involvement in TTP pathogenesis has been supposed.

Many studies highlighted complement activation in TTP by indirect signs. C3a, C5a and soluble C5b9 are markedly elevated during acute phase and normalized during remission [124,125]. High levels of these markers are associated with poor outcome [126]. In accordance with these data, therapeutic plasma exchanges strongly decrease these markers even if their level are probably lower than in atypical HUS [127]. Only the APC seems to be involved in TTP. Indeed, Tati et al. have identified C3 and C5b9 in renal cortex of TTP patients and have highlighted that TTP plasma induced release of C3- and C9-coated EMVs in vitro [128]. Furthermore, Mikes et al. report a statistic correlation between the alternative pathway C3 convertase and endothelial degranulation, which was measured by carboxyterminal pro-endothelin-1 [129]. However, a causal link between APC activation and endothelial degranulation could not be establish because of the multitude of confounding factors, i.e., factors causally linked to APC and to ECs. 

Moreover, UL-VWF is a platform for activation of the alternative pathway of complement. Tati et al. showed that administration of Shigatoxin to mice led to glomerular damage and C3 deposit only if mice were KO ADAMTS13 −/− [128]. Bettoni et al. showed that plasmas from hereditary TTP patients in acute phase induced C3 and C5b9 but not C4 deposits on HMVECs with comparable intensity of induced atypical HUS plasma deposits [125]. Remarkably, there were no more complement deposit with the adjunction of recombinant ADAMTS13. Yet, ADAMTS13 had no intrinsic inhibitory action on the alternative pathway C3 convertase formation. Turner et al. highlighted colocalization between APC components, especially C3, and UL-VWF. Thereafter, Bettoni et al. demonstrated a direct interaction between C3b and the A2 domain of VWF, leading to AP-C3/C5 convertase and C5b9 formation [125,130]. They also showed in vitro that the thrombus formation induced by TTP plasmas was corrected by restoring ADAMTS13 activity but also by complement inhibition. UL-VWF lose the ability of “normal” VWF multimers to act as cofactor of CFI, therefore losing the role of APC regulator [131]. Bettoni et al. proposed a mechanistic overview: during TTP crisis, UL-VWF pile up on the endothelial cell surface and act as platform for activation of the APC, leading to C5a and C5b9 formation that induce WPB degranulation resulting in an amplification loop. Furthermore, C5a induces shedding of TM, which downregulates coagulation cascade and indirectly inactivates C3b, as a cofactor of CFH and CFI, C3a and C5a, as a cofactor of the thrombin-activatable fibrinolysis inhibitor [27].

Recently, Zheng et al. demonstrated in a mice model that ADAMTS13 genetic deficiency had synergistic effect with APC over-activation resulting from heterozygous mutation of CFH, leading to severe TMA. Each of these genetic defects were asymptomatic when isolated [132].

P-selectin is another compound of WPBs and is expressed on activated ECs. It is a platelet and leukocyte adhesion molecule and may also serve as a platform for APC activation [133]. Thus, P-selectin may act as an amplification loop through different ways. P-selectin interacts with C3b [27,134], leading to the formation of C3a, C5a and C5b9 and the endothelial activation [135,136]. Furthermore, P-selectin participates to neutrophils recruitment, which are more cytotoxic during TTP crisis and seem to induce a complement-dependent loss of thromboresistance in vitro [137]. Finally, P-selectin is also expressed on activated platelets, which may also act as a platform for APC activation [138].

Nevertheless, Eculizumab has exceptionally been used in TTP. One case reported a very rare overlap syndrome: the patient had auto-antibodies against ADAMTS13 and against CFH [139,140]; one case was a hereditary TTP [141]; two recent cases were i-TTP with heterozygous variants of CFH, CFI and C3 [142]. These few observations do not allow us to conclude on the interest of Eculizumab in TTP.

### 5.2. Hemolysis

Intravascular hemolysis has been for long suspected to act as an amplification loop of ECs activation in TMA. Hemoglobin released induces NO consumption and produces reactive oxygen species [143], leading to endothelial dysfunction and CWP degranulation [144,145]. Hemoglobin also participates to decrease ADAMTS13 activity [146]. Furthermore, free heme has been suspected to be a critical event leading to atypical HUS when occurring on a deleterious underlying condition. Free heme induces (i) WPB degranulation in a TLR4-dependant manner [147], (ii) APC activation on ECs and (iii) APC activation in the fluid phase through facilitation of C3-C3 homophilic interaction [148,149]. We have recently shown high levels of plasmatic free heme in TTP patients. Furthermore, ECs treatment with hemopexin mildly decreased calcium influx and WPB degranulation in vitro, which supports a modest amplification role of free heme [68].

### 5.3. Nucleosomes

In response to inflammatory or infectious stimuli, neutrophils can release Neutrophils Extracellular Traps (NETs), composed of nucleosomes (DNA and histone) and of proteins, such as myeloperoxidase (MPO), neutrophil-elastase and cathepsin G. NETs have anti-infectious properties as they are “trapping” bacteria [150]. However, they also seem to have prothrombotic properties [151]. Furthermore, histones can induce consumptive thrombocytopenia in mice [152]. In 2012, Fuchs et al. showed that in patient with TMA disorders, high levels of circulating DNA and MPO were associated with low platelet counts and a low ADAMTS13 activity. These markers normalized during remission [153]. Moreover, nucleosomes are incriminated for endothelial activation in typical HUS and in allogeneic HSCT related TMA [154,155]. In this context, interleukin-8 released from endothelium may induce NETs production, which act as platform for APC [156]. As discussed above, Lysine-rich histones are used to trigger TTP in a zebrafish model. This suggests a potent mechanistic link between inflammation and TTP crisis. Moreover, Michels et al. have shown that Lysine-rich histones may act as WPB degranulation inducers by a Ca^2+^-, caspase- and charge-dependent mechanism [76]. Furthermore, ischemic damages can also lead to DNA and histones release [157], therefore acting as another prothrombotic amplification loop. Our team recently confirmed high levels of nucleosomes in plasma from acute TTP patients and a participation of these nucleosomes in iTTP patient’s plasma ECs activation [68].

## 6. Conclusions

Currently, we admit that chronic endothelial activation leads to major endothelium dysfunctions implying deregulation of hemostasis, inflammation and vascular tone. Similarly, in some circumstances, such as TMA, an acute endothelial activation can also have deleterious consequences. However, in some circumstances, such as TMA, an acute endothelial activation can also lead to deleterious consequences. We suggest that in TTP, endogenous factors, such as antibodies, cytokines, heme, histones or exogenous factors, such as virus, drug or toxin, induce an endothelial activation affecting mainly microvascular cells. This leads to a Ca^2+^-dependent WPB degranulation, resulting in UL-VWF accumulation in the microcirculation. These events may be cumulative and may be associated with amplification loops, to reach an activation level such as balance is disrupted and TMA occurs. P-selectin expression is a consequence of ECs activation, and may be a key point of pathogenesis because UL-VWF are thought to be anchored on ECs through it [158]. EMVs may also participate of amplification and systematization of the TMA process.

Despite recent therapeutic advances, mortality and morbidity during TTP remain major concerns. We have exposed clinical and experimental data supporting the “second hit hypothesis”. Furthermore, endothelium is strategically located between blood and tissues and, therefore, preferentially accessible to drugs. Its plasticity may be an interesting feature for pharmacological modulation, making endothelial cells an attractive therapeutic target [159]. Thus, we think that there is a rational basis to pharmacologically inhibit endothelial cell activation. This would be a new therapeutic axis.

## Figures and Tables

**Figure 1 jcm-12-00758-f001:**
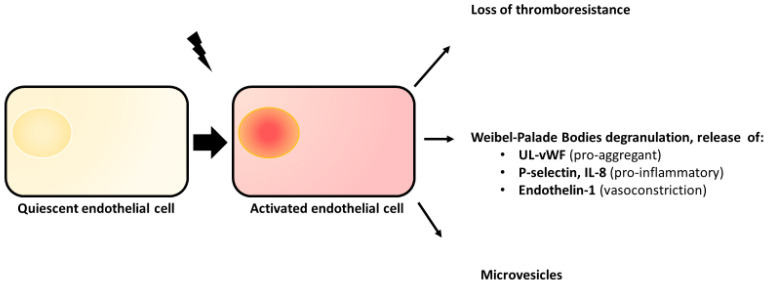
Consequences of endothelial cell activation. UL-VWF = Ultralarge von Willebrand factor; IL-8 = Interleukin-8.

**Figure 2 jcm-12-00758-f002:**
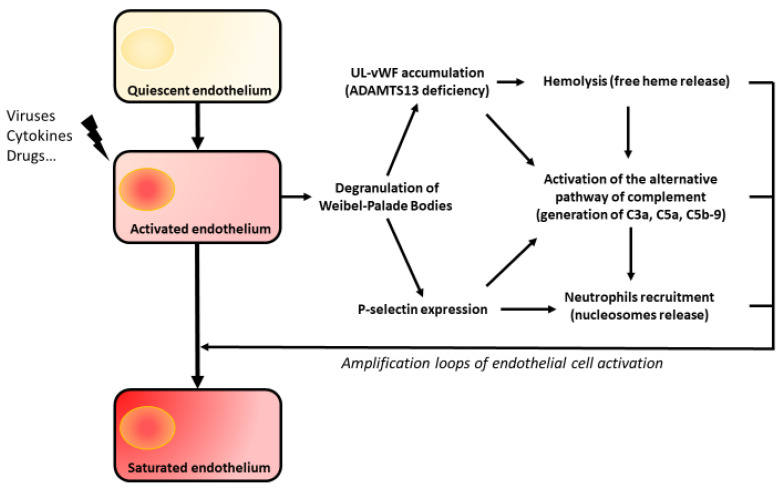
During PTT crisis, endothelium is the target of amplifications loops involving hemolysis, complement system and nucleosome UL-vWF = Ultralarge von Willebrand factor.

## Data Availability

Not applicable.

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
