# Peer review of "Is Endothelial Activation a Critical Event in Thrombotic Thrombocytopenic Purpura?"

_jcm, 2023, doi:10.3390/jcm12030758_

Round 1

Reviewer 1 Report

I completed reviewing this manuscript and I appreciate the amount of work that went into writing this text. However, I think the text does not flow easily to the reader. There is word usage especially early in the text that makes it hard to read. For example, we should state that TTP is fatal if not treated yet it is just stated as fatal period in the first page of the text. There is more text like that which needs to be revised throughout. The emphasis of this review is to make the endothelial cell as the source or site for the secondary hit in TTP. This is potentially a good area of work that will need further research. Nevertheless, this still does not explain the bone marrow suppression state recently described in iTTP that makes this a central mechanism leading to TTP presentations. In light of this, the endothelial insult would likely explain formation of diffuse systemic microthrombi as mentioned by the authors but it does not explain how the bone marrow does not seem to respond with an increase in platelets during disease (PMID 35617796, 33290885). Mechanistically, there may be a multilayered presentation in which as platelets get trapped in the increasingly prothrombotic periphery, the marrow fails to respond with an increase output of platelets in the form of young platelets making the patient increasingly thrombocytopenic.

Author Response

Reviewer #1

I completed reviewing this manuscript and I appreciate the amount of work that went into writing this text.

We warmly thank the reviewer for his encouragement

However, I think the text does not flow easily to the reader. There is word usage especially early in the text that makes it hard to read. For example, we should state that TTP is fatal if not treated yet it is just stated as fatal period in the first page of the text. There is more text like that which needs to be revised throughout.

We are in complete agreement with the reviewer. A careful proofreading and many corrections were made by a native speaker to the manuscript. We hope that the corrections made will satisfy the reviewer.

The emphasis of this review is to make the endothelial cell as the source or site for the secondary hit in TTP. This is potentially a good area of work that will need further research. Nevertheless, this still does not explain the bone marrow suppression state recently described in iTTP that makes this a central mechanism leading to TTP presentations. In light of this, the endothelial insult would likely explain formation of diffuse systemic microthrombi as mentioned by the authors but it does not explain how the bone marrow does not seem to respond with an increase in platelets during disease (PMID 35617796, 33290885). Mechanistically, there may be a multilayered presentation in which as platelets get trapped in the increasingly prothrombotic periphery, the marrow fails to respond with an increase output of platelets in the form of young platelets making the patient increasingly thrombocytopenic.

We thank the reviewer for this important element. We have added this aspect of pathophysiology at the end of the introduction, and we have cited the 2 corresponding articles.

We hope that the answers in this manuscript and modifications we brought in the reviewed manuscript will be satisfying.

Sincerely yours,

Dr Raphael CAUCHOIS, MD

Reviewer 2 Report

Congratulations for this nice work. 

I suggest the following in the introduction section: Thrombotic thrombocytopenic purpura (TTP) is a rare and severe condition belonging to the thrombotic microangiopathies disorders. It is characterized by uncontrolled platelet aggregation and adhesion which will form microthrombi resulting in the clinical syndrome of TTP. It can be fatal without prompt diagnosis and appropriate treatment. TTP results from deficiency of ADAMTS13, a vWF cleavage protease enzyme, leading to impaired processing and accumulation of ultra large vWF multimers and formation of microthrombi. ADAMTS13 deficiency can be inherited as in congenital TTP (Upshaw-Schulman Syndrome) or more commonly acquired due to autoantibodies against ADAMTS13 which is usually called immun e mediated TTP.

In the section 4. What are the suspected triggers for the 2nd hit hypothesis? I suggest a brief description of the possible pathogenesis of immune-mediated TTP, e.g; HLA class II (HLA DRB1*04 as protective and HLA DRB1*11 and DQB1*03 as predisposing) and autoimmuninty.

Author Response

Reviewer #2

Congratulations for this nice work. 

We warmly thank the reviewer for his encouragement

I suggest the following in the introduction section: Thrombotic thrombocytopenic purpura (TTP) is a rare and severe condition belonging to the thrombotic microangiopathies disorders. It is characterized by uncontrolled platelet aggregation and adhesion which will form microthrombi resulting in the clinical syndrome of TTP. It can be fatal without prompt diagnosis and appropriate treatment. TTP results from deficiency of ADAMTS13, a vWF cleavage protease enzyme, leading to impaired processing and accumulation of ultra large vWF multimers and formation of microthrombi. ADAMTS13 deficiency can be inherited as in congenital TTP (Upshaw-Schulman Syndrome) or more commonly acquired due to autoantibodies against ADAMTS13 which is usually called immun e mediated TTP.

We appreciate the wording proposed by the reviewer. We have taken the essence of the proposed syntax.

In the section 4. What are the suspected triggers for the 2nd hit hypothesis? I suggest a brief description of the possible pathogenesis of immune-mediated TTP, e.g; HLA class II (HLA DRB1*04 as protective and HLA DRB1*11 and DQB1*03 as predisposing) and autoimmuninty.

We thank the reviewer for this advice. We have added a small section regarding the pathophysiology of autoimmunity in TTP.

We hope that the answers in this manuscript and modifications we brought in the reviewed manuscript will be satisfying.

Sincerely yours,

Dr Raphael CAUCHOIS, MD

Round 2

Reviewer 1 Report

As I mentioned in my original report. The paper was not well conceived. It had no flow and it consisted of paragraphs of text without cohesiveness. Second, the references included were not thorough and excluded significant up to date publications which mentioned the changing landscaping of how we see the pathogenesis of TTP. Third, the text needed to be fully rewritten based on new areas in the field which the authors chose not to cite, including potential second hits needed for TTP pathology. Fourth, far superior reviews have been published which go over the most recent findings within the last year which this manuscript fails to include.
